# Enhanced Energy Storage Using Pin-Fins in a Thermohydraulic System in the Presence of Phase Change Material

Mohamad Ziad Saghir

Department of Mechanical and Industrial Engineering, Toronto Metropolitan University, Toronto, ON M5B 2K3, Canada; zsaghir@ryerson.ca

**Abstract:** Energy storage has been an essential topic in thermal management. With the low conductivity of phase change material, the effort is to propose the best mechanism for heat transfer. In the present paper, pin-fins are used in the hydraulic system to transfer the heat coming from wastewater management into phase change material. Different flow rates have been tested, and it was found that pin-fins can create mixing in the flow chamber allowing a large convective heat flux to move heat into the phase change material. In the present design, it was found that natural convection assists in heat transfer. Additional findings suggested that the pin-fins height influence the heat transfer process. In the current configuration, 5 mm in height pin-fins demonstrated the best heat transfer when compared to pin-fins varying from 1 mm to 6 mm, respectively.

**Keywords:** phase change material; chevrons; pin-fins; energy storage; pin-fins height

## 1. Introduction

The utilization of renewable energy and waste heat is becoming increasingly important, and this can be attributed to rising energy demand and climate targets for decarbonization of the energy sector. Due to their intermittent character, from temporal and geographical disparities in their availability and demand, energy storage technologies are required to apply such sustainable energy resources. Thermal Energy Storage (TES) systems capture and recover energy by changing the internal energy of thermal materials. This is widely used due to its simplicity and affordability but does fall short of low energy density and the requirement of a large volume of storage media. Phase change materials (PCM) are a new development that can perform better and more efficiently. This is because they store and release energy through latent heat of phase transition, providing more significant energy density and many other benefits. There are three categories for PCMs: Organic—paraffin and non-paraffin, inorganic, and eutectic. The main features that are looked for in organic PCMs are appropriate fusion temperature range, sizeable latent heat, and chemical stability. There are a few drawbacks with organic PCMs, including low thermal conductivity, which significantly impacts performance for TES. The low thermal conductivity restricts the heat transfer rate within the PCM, reducing the system's charging and discharging power. Pin-fins, or expanded metal surfaces, appear to be the most practical and straightforward option.

However, finding an efficient energy storage system for renewable energy is of great importance. The use of thermal energy storage (TES) to collect, store, and conserve energy for short- or long-term use in modern energy generation systems has become a demand [1]. TES systems can be broken into three categories: latent heat, thermochemical energy storage (TCES), and sensible heat. In sensible energy TES, the storage media could be thermal oil, concrete, molten salt, stone, metal, ground, and water. In contrast, latent heat TES systems use phase change materials (PCMs) to store thermal energy, such as organic (paraffin and non-paraffin), inorganic (salt hydrates, molten salts, and metals), and eutectic (organics, inorganics, or both) materials. The main drawbacks of sensible and latent heat are their heat loss, short-term storage, and low thermal conductivity [1–5].

The concept of TES includes storing heat or cold based on the temperature range of the thermal battery. The thermal system's efficiency increases from below 50% up to 70 to 100% when using the TES method for generating heat during discharging [6,7].

Renewable energy is widely used as an energy source, but unreliable and low-density energy storage technologies remain a significant challenge [8]. Therefore, an efficient and environmentally friendly energy storage solution is essential, and thermal energy storage (TES) is widely promoted. Currently, the three mainstream thermal energy storage solutions are water tank heat storage, phase change heat storage, and thermochemical heat storage [9]. The thermochemical heat system (THS) is cutting-edge technology. Studies have shown that THS has many advantages over other thermal energy storage systems, such as tank storage, phase change heat storage, and others [10–14].

Thermal energy storage performance of paraffin in a novel tube-in-shell system, using phase change materials (PCMs) to store thermal energy, is an efficient method. PCMs are a viable thermal management technology for intermittent heat loads because they hold thermal energy as latent heat [15–17]. When looking at these systems, the shape of the container and structure is also essential [18,19].

Dealing more with PCMs, in a comprehensive parametric study, Moon et al. [20] used carbon foam structures saturated with PCMS in thermal management. Jian et al. [21] proposed a high-power density thermal energy storage using additively manufactured heat exchangers and phase change material, the use of phase change materials (PCMs) to store thermal energy as an effective method is discussed.

Ismail et al. [22] proposed polymer composite materials as an indispensable part of electronic products. Radiation fins appear in many heat transfer mechanisms, and the fins can receive and transfer heat well. In Ismail et al. [23] 's study, the enthalpy formula and the controlled volume method were used to calculate thermal efficiency. Finite-difference approximations and alternating directions schemes are used to discretize the fundamental equations and associated boundary and initial conditions. The effects of the number of fins and the length of the fins were verified through multiple experiments. The study shows that the annular space, the radial length of the fins, and the number of fins significantly influence the solidification mass fraction and the whole transformation time. Hoseinzadeh et al. [24] studied how heat flows in porous fins. The Darcy model is used to simulate heat transfer in such porous media. Assuming the fins are one-dimensional and uniform, the flow is laminar, and the heat generated is a linear function of temperature. They used three different analytical methods to obtain the temperature distribution. Amongst them are the collocation method (CM) and the numerical method. In addition, the homotopy perturbation method (HPM) was used to verify the solutions. Xiaohu et al. [25] showed that annular fins have better heat dissipation efficiency. By inserting annular fins in the PCM, they could minimize the total melt time by 65%. Compared with continuous fins, adjusting the fluid flow in microchannels with segmented inclined fins can effectively enhance heat transfer efficiency [26].

Finned tubes are widely used in the waste heat recovery system of internal combustion engines. The exhaust gas of an internal combustion engine takes approximately 30% of the combustion energy, and if this heat is not used correctly, it is considered wasted energy. The study found that nearly 10–15% of the fuel power is stored in the combined storage system as heat, available for suitable applications at reasonably high temperatures [27]. Abidi et al. [28] showed that the thermal conductivity of PCM directly affects the cost and size of the cooling system. Due to the low thermal conductivity, the structure of the traditional heat dissipation grill must be improved, which directly increases the initial design cost.

In the present paper, an attempt is made to determine whether using pin-fins can accelerate heat transfer to the phase change material. The chevron shape of the channel was used in the present study. The pin-fins forming the chevron shape channels varied from 1 mm in height to 6 mm in height. Section 2 presents the current problem under investigation, followed by Section 3, which is devoted to the numerical simulation formulation. Section 4 focuses on model validation with existing experimental data, and Section 5 presents the results and discussion, followed by the conclusion in Section 6.

## 2. Problem Statement

Yazdani et al. [29] conducted an accurate experimental study on energy storage in phase change material from waste heat recovery. A metallic grid is introduced in their design because the phase change material has low conductivity. The void space in the metallic grid contained phase change material that is paraffine. They demonstrated that this approach led to fast energy absorption to the phase change material. Saghir [30] showed that for a system containing pin-fins, heat dissipates faster whether one uses water or any other fluid, such as nanofluid. The pin-fins helped to enhance the heat extraction. In addition, Saghir [30] demonstrated that the height of pin-fins plays a vital role in heat enhancement. The pin-fins distribution used by Saghir [31] was arranged as chevrons or as channel walls or even as wavy channel walls. In addition, the wavy channel allows the heat to circulate longer; thus, the fluid absorbs more heat. Interestingly, the findings are that the presence of pin fins does not create a considerable pressure drop and creates a convective flow aimed at mixing. At the same time, we remain in the laminar regime.

Thus, in the current paper, we intend to duplicate the Yazdani model without using a metallic grid but instead introduce pin-fins in the flow region. The intention is to determine whether pin-fins can accelerate the heat dissipation toward the phase change material in a short period. Thus, if successful, one then does not need a metallic grid. Figure 1a presents the model under investigation. The model consisted of a block where the waste warm water circulates within the pin-fins and is sandwiched between two blocks of phase change material. The three blocks have a square base of 37.5 mm by 37.5 mm and a height of 12.7 mm. Figure 1b shows the distribution of the pin-fins in the middle block. The flow enters from the inlet, which is a cylinder with a diameter of 8 mm. The wastewater in our study enters from the inlet, circulates between the pin-fins, and exits from the opposite end. The inlet flow velocity is 0.05 m/s, 0.01 m/s and 0.015 m/s, corresponding to the Reynolds number of 50, 100 and 150, respectively. The advantage of these pin-fins is to force the flow to circulate between the obstacle, and water deflects to the top and bottom surfaces of the block. This flow mixing accelerates the heat transfer to the phase change material, while remaining in the laminar regime. The phase change material used in our simulation is paraffine, and the physical properties are presented in Table 1.

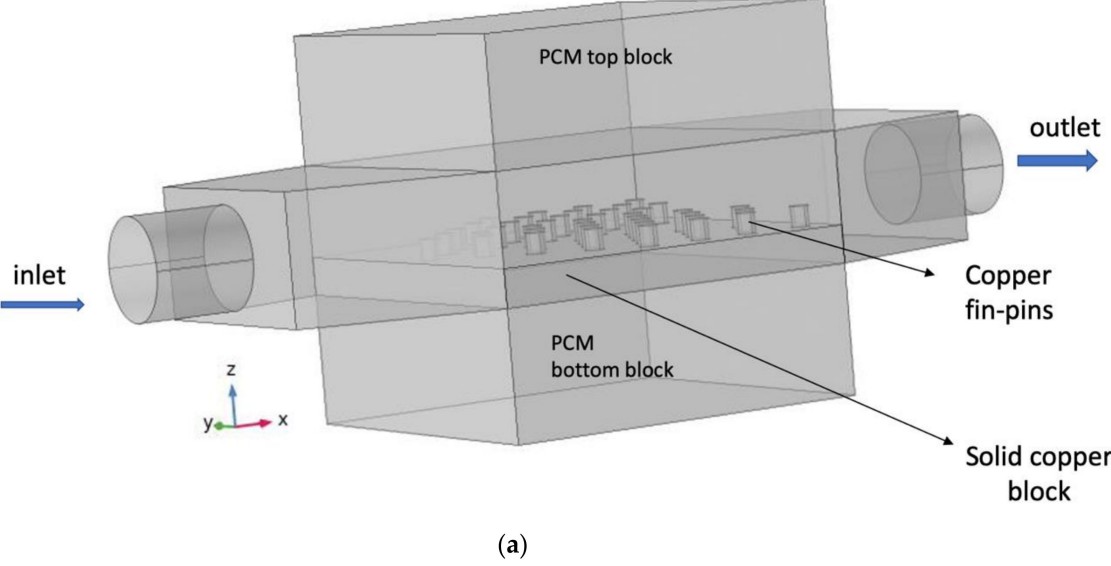

(**a**)

**Figure 1.** *Cont.*

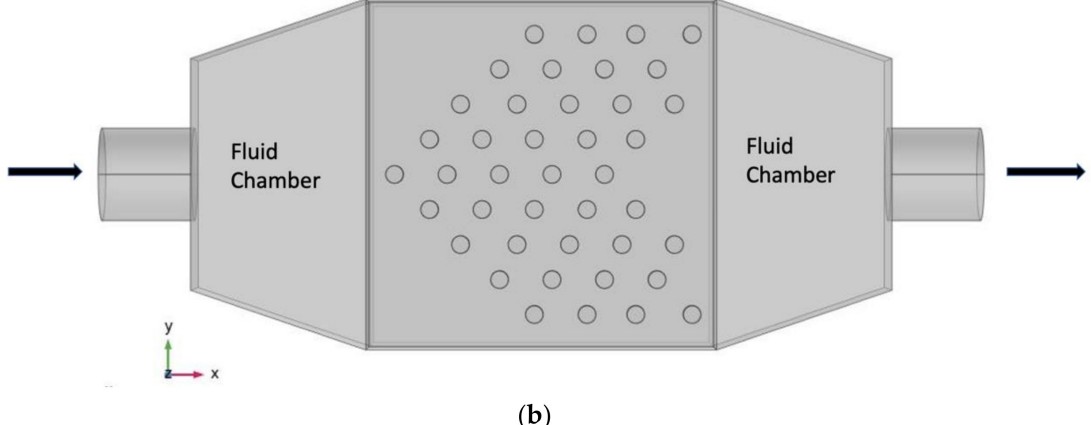

**(b)**

**Figure 1.** Model with phase change material.

**Table 1.** Paraffins' physical properties [29].

| $\rho_1$ (kg/m$^3$) | $\rho_2$ (kg/m$^3$) | $Cp_1$ (J/kg/K) | $Cp_2$ (J/kg/k) | $k_1$ (W/m/s) | $k_2$ (W/m/s) | $L_{1\rightarrow 2}$ (KJ/kg) |
|---|---|---|---|---|---|---|
| 900 | 770 | 2400 | 1900 | 0.35 | 0.15 | 210 |

## 3. Finite Element Model and Boundary Conditions

Based on the model presented in Figure 1a, the full Navier Stocks equations were solved for the fluid region and the energy equation for the entire model. The flow is assumed Newtonian, and the flow rate is low to remain in the laminar regime. The fluid flow equations in the three-dimensional form are as follows:

x-direction

$$\rho\left[\frac{\partial u}{\partial \tau} + u\frac{\partial u}{\partial x} + v\frac{\partial u}{\partial y} + w\frac{\partial u}{\partial z}\right] = -\frac{\partial p}{\partial x} + \mu\left[\frac{\partial^2 u}{\partial x^2} + \frac{\partial^2 u}{\partial y^2} + \frac{\partial^2 u}{\partial z^2}\right] \tag{1}$$

y-direction

$$\rho\left[\frac{\partial v}{\partial \tau} + u\frac{\partial v}{\partial x} + v\frac{\partial v}{\partial y} + w\frac{\partial v}{\partial z}\right] = -\frac{\partial p}{\partial y} + \mu\left[\frac{\partial^2 v}{\partial x^2} + \frac{\partial^2 v}{\partial y^2} + \frac{\partial^2 v}{\partial z^2}\right] \tag{2}$$

z-direction

$$\rho\left[\frac{\partial w}{\partial \tau} + u\frac{\partial w}{\partial x} + v\frac{\partial w}{\partial y} + w\frac{\partial w}{\partial z}\right] = -\frac{\partial p}{\partial z} + \mu\left[\frac{\partial^2 w}{\partial x^2} + \frac{\partial^2 w}{\partial y^2} + \frac{\partial^2 w}{\partial z^2}\right] - \rho g \tag{3}$$

In Equations (1)–(3), the velocities u, v, and w are in the directions of x, y, and z, respectively. The gravity vector is in the z-direction. The fluid density is $\rho$ in kg/m$^3$, and the dynamic viscosity is $\mu$ in kg/m/s.

The energy equation for the fluid portion is as follows:

$$\rho Cp\left[\frac{\partial T}{\partial \tau} + u\frac{\partial T}{\partial x} + v\frac{\partial T}{\partial y} + w\frac{\partial T}{\partial z}\right] = k\left[\frac{\partial^2 T}{\partial x^2} + \frac{\partial^2 T}{\partial y^2} + \frac{\partial^2 T}{\partial z^2}\right] \tag{4}$$

The specific heat capacity is Cp in J/kg/K, and T is the temperature in degrees K for the entire model. Here, the conductivity of the flow is k in W/m/K. The temperature in the solid part of the model is determined by solving the energy formulation shown in Equation (4) without the convective term.

The phase change model's density is:

$$\rho_{pcm} = \varnothing_1\rho_1 + \varnothing_2\rho_2 \tag{5}$$

where $\varnothing_1$ and $\varnothing_2$ are the molten fraction phase and the solid phase of the material, respectively. The density of the phase change material at each state is $\rho_1$(molten) and $\rho_2$ (solid) in $kg/m^3$. Similarly, the specific heat of the phase change material is:

$$Cp_{pcm} = \frac{1}{\rho_{pcm}}(\varnothing_1\rho_1 Cp_1 + \varnothing_2\rho_2 Cp_2) + L_{1\rightarrow 2}\frac{\partial\alpha_{pcm}}{\partial T} \qquad (6)$$

Here $\alpha_{pcm}$ is defined as:

$$\alpha_{pcm} = \frac{\varnothing_2\rho_2 - \varnothing_1\rho_1}{2(\varnothing_2\rho_2 + \varnothing_1\rho_1)} \qquad (7)$$

The latent heat is known as $L_{1\rightarrow 2}$ in J/kg, and the conductivity of the phase change material is:

$$k_{pcm} = \varnothing_1 k_1 + \varnothing_2 k_2 \qquad (8)$$

where $k_1$ and $k_2$ correspond to the conductivity of the solid and liquid phase change material in W/m/K. It is also apparent that the sum of the two fraction phases equals 1.

The boundary conditions applied to this model are as follows:

At the inlet

The inlet velocity u is set equal to 0.005 m/s or 0.01 m/s or 0.015 m/s, which correspond to a Reynolds number of 50, 100 and 150, respectively. The inlet temperature T is set equal to 65 degrees Celsius. Thus,

$$u = 0.005 \text{ m/s or } \quad 0.01 \text{ m/s or } 0.015 \text{ m/s} \qquad (9)$$

$$Tin = 65 \text{ degrees Celsius} \qquad (10)$$

At the outlet

The free boundary is applied, and the stresses are equal to zero.

All external surfaces are assumed insulated, so there are no heat losses. This assumption is difficult to make as a heat leak is occurring. Thus,

$$\frac{\partial T}{\partial n} = 0 \qquad (11)$$

*Mesh Sensitivity and Convergence Criteria*

To ensure the model has the right mesh size, mesh sensitivity was conducted, and as shown in Table 2, the normal mesh was found to be appropriate for our model. Figure 2 presents the mesh used in the model.

Different approaches exist in COMSOL [32] to tackle the convergence criteria. In this model, the default solver used was the segregated method. The convergence criteria are clearly explained in the COMSOL manual. In a short summary, the convergence criteria were set as follows: at every iteration, the average relative error of u, v, w, p and T were computed. In total, 124,125 elements were used in the model. These were obtained using the following relation:

$$R_c = \frac{1}{n \cdot m}\sum_{i=1}^{i=m}\sum_{j=1}^{j=n}\left|\frac{\left(F_{i,j}^{s+1} - F_{i,j}^{s}\right)}{F_{i,j}^{s+1}}\right| \qquad (12)$$

where F represents one of the unknowns, viz., u, v, w, p, or T, where s is the iteration number and (i, j) represents the coordinates on the grid. Convergence is reached if $R_c$ for all the unknowns is below $1 \times 10^{-6}$ in two successive iterations. For further information on the detailed solution method, the reader is referred to the COMSOL software manual [26].

**Table 2.** Mesh sensitivity analysis.

| Mesh Size | Number of Elements | Nusselt Number |
| --- | --- | --- |
| Extra coarse | 17,888 elements | 5 |
| Coarser | 30,763 elements | 4.7 |
| Coarse | 68,890 elements | 4.4 |
| Normal | 124,125 elements | 4.2 |

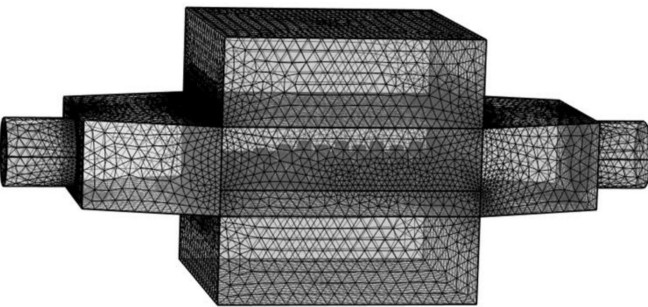

**Figure 2.** Finite element mesh.

## 4. Model Validation with Experimental Data

Ambreen et al. [33] conducted an experimental measurement of the Nusselt number using nanofluid to investigate pin-fins' importance in heat removal. The mixture consists of MXene nanoparticles having a concentration of 0.013% and 0.027%vol in distilled water. As shown in Figure 3a, the flow enters perpendicular to the pin-fins base and circulates around the pin-fins before exiting from the opposite side of the system. The bottom copper plate is subject to a heat flux. The pin-fins have a diameter of 1.5 mm and a height of 7.5 mm. They are made of copper as well as the base plate, which is subject to heating.

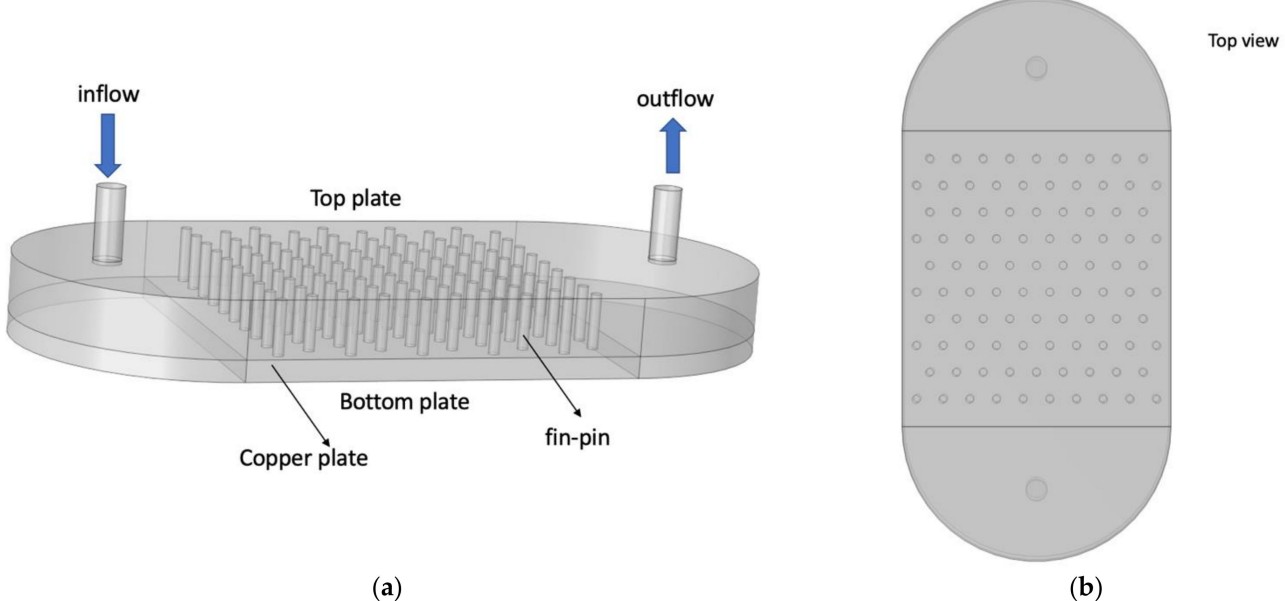

(**a**)                    (**b**)

**Figure 3.** Experimental [30] and current numerical setup.

The temperature measurements are done at the base of the pin-fins, and the nanofluid Nusselt number is measured experimentally. The flow rate represented by the Reynolds number varied between 600 and 2000. As shown in Figure 3b, the pin-fins have a particular, uniform arrangement in the diagonal position. The problem was solved numerically to

check the accuracy of the numerical model used in this analysis. Figure 4 displays the average nanofluid Nusselt number as a function of the Reynolds number for a nanofluid having a 0.013% MXene concentration. As shown, the slope of the Nusselt number gradient is similar between the two methods, and the data shows an excellent agreement. The definition of Nusselt number for nanofluid could be found in reference [30].

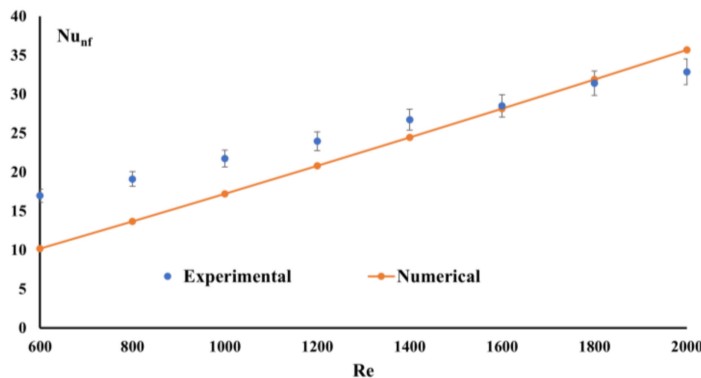

**Figure 4.** Average nanofluid Nusselt number versus the flow rate.

In the numerical approach, the outside surfaces are assumed insulated, but experimentally it is evident that some heat leaked to the outside. Other experiment errors may occur from heating and the flow rate accuracy, which may contribute to this difference in results. However, the numerical code predicts very well the experimental results. The numerical model used in the validation is very similar to the proposed. The pin-fins arrangement is different, and two-phase change material blocks are added. In addition, the location of the inlet and outlet pipe are different. A detailed description can be found in Figure 1.

## 5. Results and Discussions

This paper attempts to introduce pin-fins in the flow region, aiming at enhancing heat transfer to the phase change material. Pin-fins have been used in many engineering applications in the aerospace industry, as mentioned in the introduction. As mentioned earlier, Saghir [31] demonstrated the usefulness of introducing a pin-fins channel for heat extraction. On the one hand, the pin-fins do not present significant pressure drop; on the other hand, they are demonstrated to be an excellent heat extraction tool. Because phase change material has low thermal conductivity, it is essential to determine the optimum design for heat storage. What was also demonstrated earlier by Saghir [31] is that the height of the pin-fins may have a direct effect on heat extraction, thus another parameter to be investigated in the present paper. In recent work by Saghir and Ghalayini [32], pin-fins forming Chevron shapes were used. Their study demonstrated that this type of channel is effective in heat removal [32]. This work's uniqueness and novelty is the introduction of a new tool for heat storage without the need to trap the phase change material in a conductive grid.

### 5.1. Effectiveness of Using Pin-Fins for Enhanced Heat Transfer

In a previous study by Saghir [31], pin-fins have proven to be an effective tool to extract more heat into the circulating fluid. However, it was also demonstrated that this heat enhancement would not affect the pressure drop. Thus, the performance enhancement criterion remains very high. To display the effectiveness of pin-fins, the model in Figure 1a is studied numerically without the presence of phase change material [31]. The model was heated from the bottom plate, and a water flow was circulating through the pin fins. Figure 5 shows the performance with the presence of pin-fins compared to the case of no pin-fins. In the present configuration, the pin fins arrangement is identical to the current structure of chevron-like and removed to demonstrate the effect of no pin-fins. The pin-fins dimension and height are similar to the present configurations. As shown, the presence

of pin-fins can enhance heat removal. The uniqueness of the existence of pin-fins is the creation of mixing flow, thus increasing the convective heating while remaining in the laminar regime.

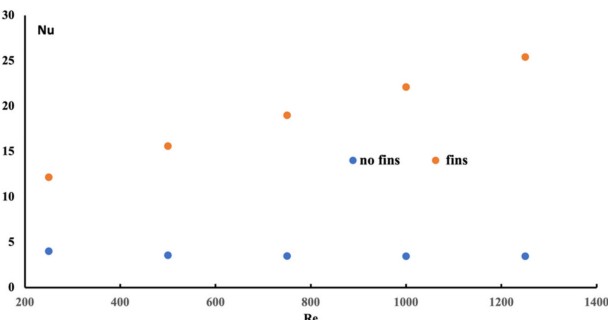

**Figure 5.** Effectiveness of using pin-fins along the flow path.

With two validations presented, the model is modified, as shown in Figure 1a. The effectiveness of using this design for heat storage in the presence of phase change material is investigated.

### 5.2. Phase Change Material in Block

In the current study, the water enters from the inlet at a temperature of 65 degrees Celsius and dissipates the heat in the entire system. The flow circulates between the chevron-distributed pin-fins and exits from the opposite end, as shown in Figure 1. This process laps for 700 s, and then the inlet temperature drops to 20 degrees Celsius. Thus, all the heat will be extracted from the entire system, including the phase change material. This process is called the charging and discharging of the system. Figure 6 displays the imposed temperature variation at the inlet and the obtained temperature at the outlet for a typical condition as a function of time. At the inlet, a step function temperature is imposed, and a nonlinear temperature is obtained at the outlet. This inlet temperature profile will be used for the entire case presented in Figure 6.

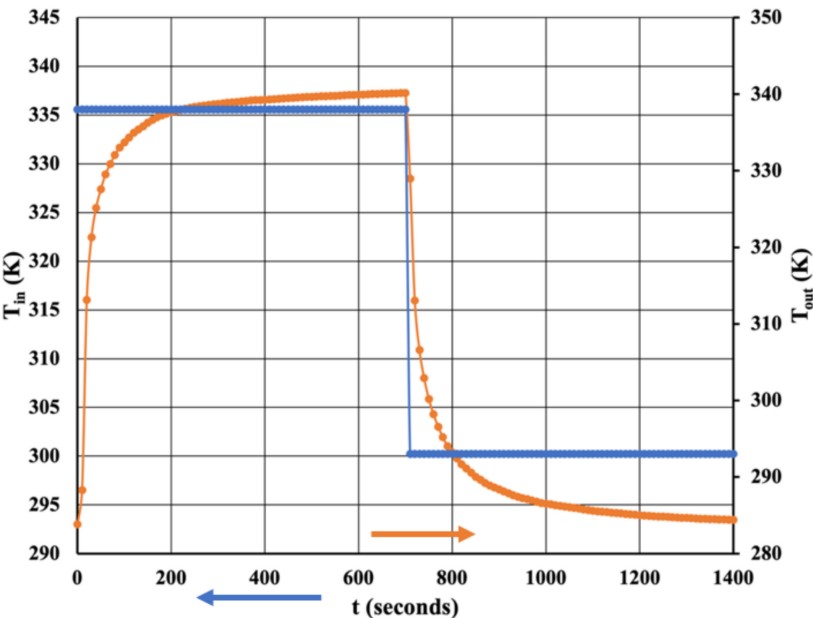

**Figure 6.** Applied inlet temperature with time.

Two critical parameters will be evaluated during the charging and discharging process. These are the energy power identified as P and the thermal energy storage capacity determined as Q. Energy power is defined as:

$$P = \dot{m}Cp\Delta T \tag{13}$$

In this equation, $\dot{m}$ is the mass flow rate of the water circulating in the system in kg/s, Cp is the water-specific heat capacity in J/kg/K, and finally, $\Delta T$ is the temperature difference between the inlet and the outlet. On the other hand, the thermal energy storage Q is presented as follows:

$$Q = \dot{m} \times \left[ Cp_1 (T_{melt} - T_{in}) \right] + \Delta L_{1-2} + Cp_s (T_{out} - T_{melt}) \right] \tag{14}$$

where $Cp_l$ is the specific heat capacity at the molten PCM phase in J/kg/s and $Cp_s$ is the specific heat capacity at the solid phase in J/kg/s and $\dot{m}$ is the mass flow rate in kg/s. The melting temperature of paraffin in degrees kelvin is $T_{melt}$ and $\Delta L_{1-2}$ is the latent heat in J/kg.

Figure 7 presents the energy power P and the thermal energy storage capacity Q as a function of time for three different Reynolds numbers. In all cases, the variation profile is identical. For the energy power P, one observes considerable energy generated and then drops with time until it reaches a minimum value of 700 s. The process is repeated with opposite variation but in the discharging mode. In the second segment, the inlet temperature drops to room temperature, and the outlet temperature starts decreasing, as discussed in Figure 6. This corresponds to reducing the energy power from the system by discharging the heat accumulated.

On the other hand, the thermal energy storage increases as the inlet temperature is approximately 65 degrees aiming at storing the energy in the phase change material. After 700 s, the stored energy is reversed. Noticeably, as the Reynolds number increases, the energy stored capacity increases accordingly. After a certain period, the energy stored is less pronounced; thus, 700 s is a good indicator of the essential part of energy harvesting. At 700 s, a singularity point occurs when the inlet temperature drops to 20 degrees Celsius, and the system switches from charging to discharging. To further demonstrate the importance of designing pin-fins in such a system, Table 3 presents the total heat stored by the phase change material during the charging period. The data is shown for the two cases of pin-fins and without pin-fins. Remember that the pin height in the present investigation is set equal to 2 mm. The model is repeated in the absence of pin-fins to demonstrate the effectiveness of using pin-fins as a simple mechanism for heat storage. As one may notice, pin-fins allow for more heat storage.

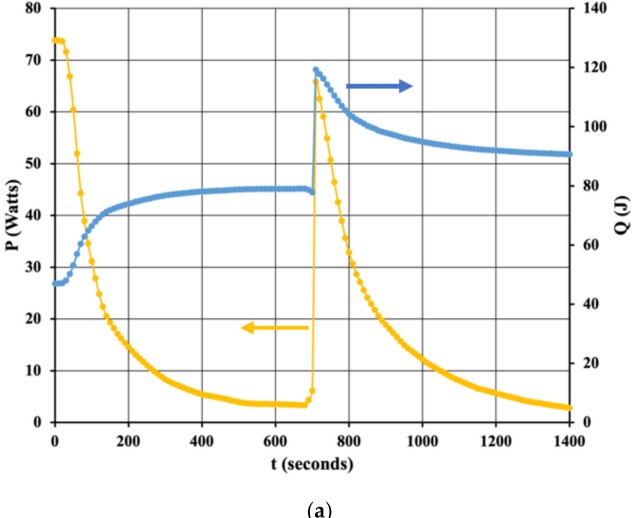

(a)

**Figure 7.** *Cont.*

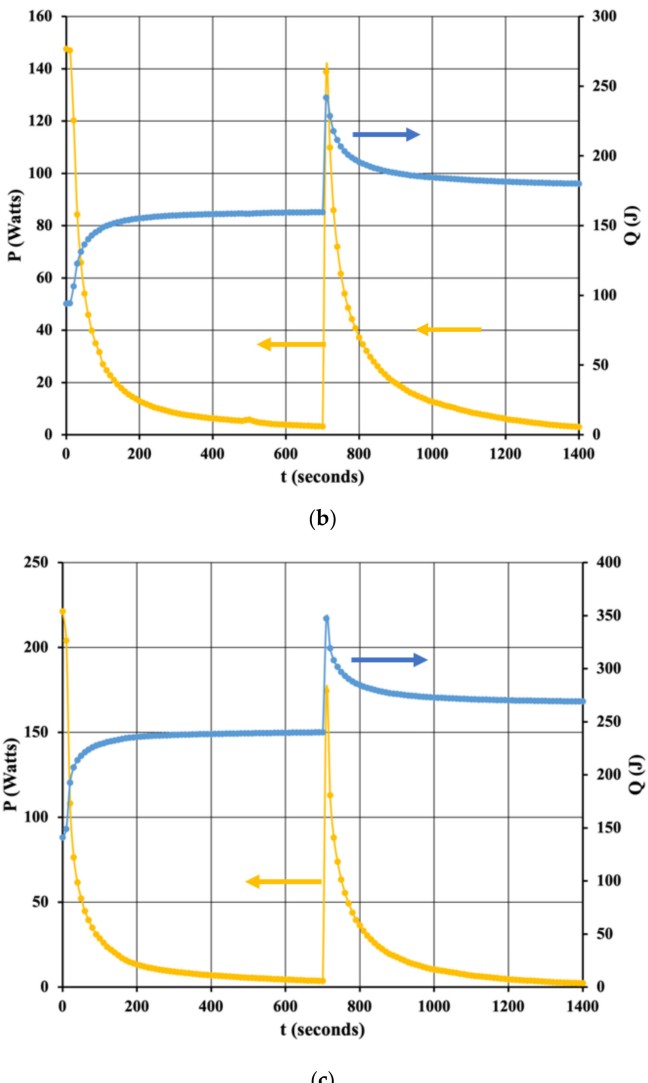

(b)

(c)

**Figure 7.** The energy charge P and discharge power Q as a function of Reynolds number and time in the presence of pin-fins (**a**) Re = 50, (**b**) Re = 100, (**c**) Re = 150.

**Table 3.** Total energy stored Q in phase change material.

| Reynolds Number (Re) | Energy Stored (J) (With Pin-Fins) | Energy Stored in (J) (Without Pin-Fins) |
|---|---|---|
| 50 | 5213 | 5210 |
| 100 | 10,854 | 10,778 |
| 150 | 16,511 | 16,465 |

From Table 3, one may notice an increase in heat storage in the presence of pin-fins.

Figures 8 and 9 present the temperature variation in the middle of the model, starting from the bottom PCM block to the top PCM block. Thus, the variation is along the z direction by maintaining the x and y coordinate fixed at (x,y) equal to (0.001875-m,0-m), respectively. Figure 7 presents the case in the presence of the pin-fins. The straight red line corresponds to the melting temperature of the PCM under study. In the presence of gravity, one may notice that the bottom PCM block is heated from the top, and the top PCM block is heated from the bottom. Thus, one expects a pure diffusion mechanism for the bottom PCM block and a natural convection for the top PCM block. The dotted temperature

line corresponds to the temperature variation during discharge, and the complete lines correspond to the temperature during the charging period. As shown in Figure 8a, at a low Reynolds number, most of the PCM is molten at the top block, and the block is in a mushy condition at the bottom. Natural convection indeed helped in distributing the heat in the top block. As the flow rate represented by the Reynolds number increases further, the melting PCM increases, and thus the top block is mainly molten. However, in the bottom block, no molten PCM is observed. The high flow rate can achieve strong natural convection, and energy is well stored in the PCM.

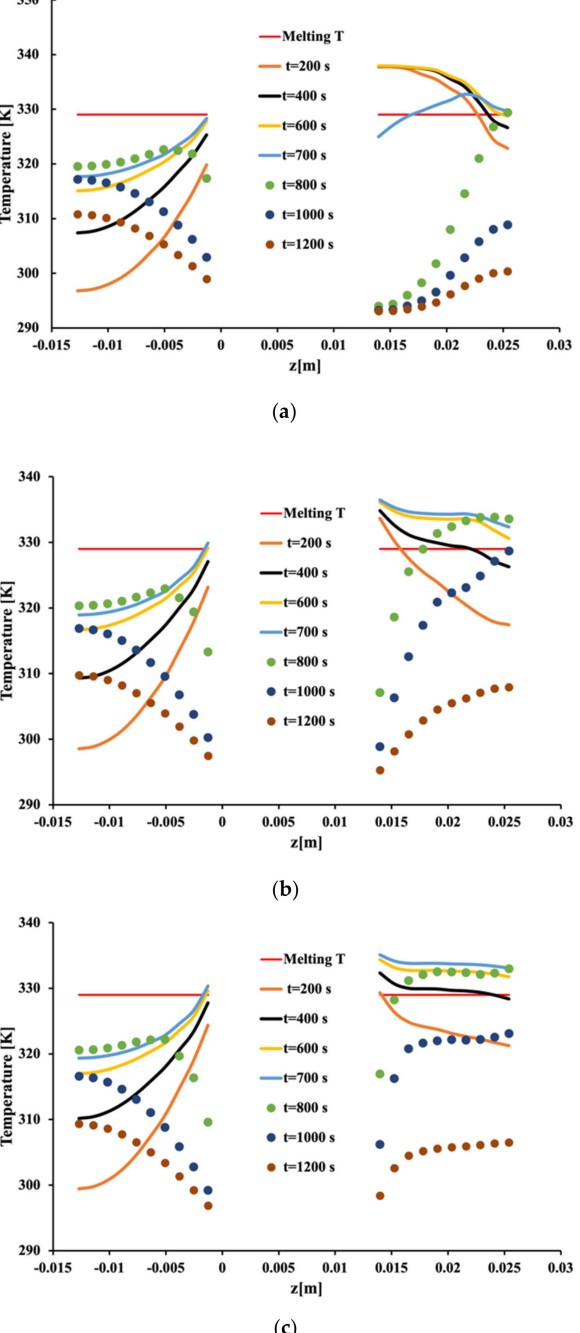

**Figure 8.** Temperature distribution along the z-axis in the middle of the phase change material. (With pin-fins, (**a**) Re = 50, (**b**) Re = 100, (**c**) Re = 150).

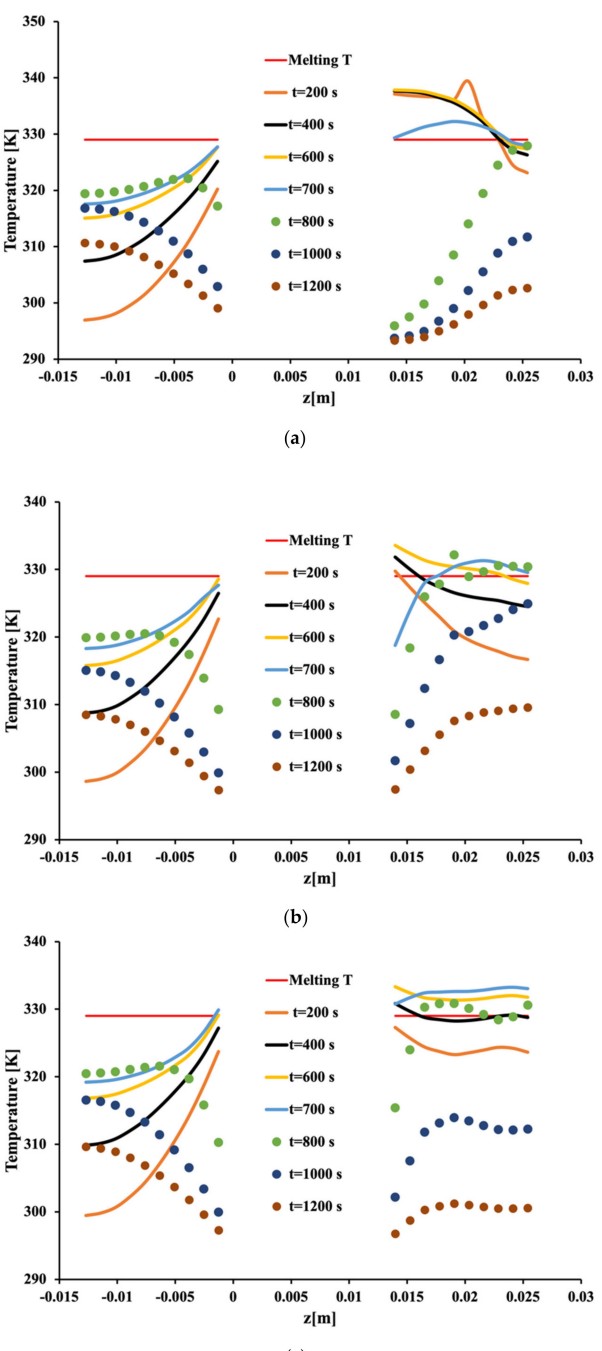

**Figure 9.** Temperature distribution along the z-axis in the middle of the phase change material. (With no pin-fins, (**a**) Re = 50, (**b**) Re = 100, (**c**) Re = 150).

Figure 8 presents the identical case and identical conditions but in the absence of pin-fins. Regardless of whether pin-fins are inserted, the bottom PCM remains in the diffusion regime; thus, a slow heating process occurs. However, looking at the temperature distribution, one may notice that the amount of heat transmitted to the PCM is much less than in the case of the presence of pin-fins. This finding will be further examined in this paper.

Another way to observe the temperature variation in the middle section of the system is to display the iso-temperature surfaces in the middle of the model, as shown in Figures 10 and 11. In Figure 10, the results are displayed at the end of the charging time at t = 700 s. Figure 10a shows the case when the Reynolds number equals 50. As seen, a natural convection is present at the top PCM block and the pure diffusion mechanism at the bottom PCM block. A convective pattern exists where natural convection exists, and uniform

temperature distribution is in the diffusion part. As the Reynolds number increases, one may notice an increase in heat transportation, as shown in Figure 10b,c. The rotating cells at the corner of the PCM block are reduced as the Reynolds number increases. During the discharge period, heat removal is observed, as shown in Figure 10. One may notice that the heat accumulated at the PCM corner takes a longer time to be released due to the low thermal conductivity of the PCM. At a time of t = 1400 s, some remaining heat trapped at the corner of the phase change material is clearly shown. However, the pin-fins were able to assist in removing heat as well as adding heat, whether it was in the charging or discharging period.

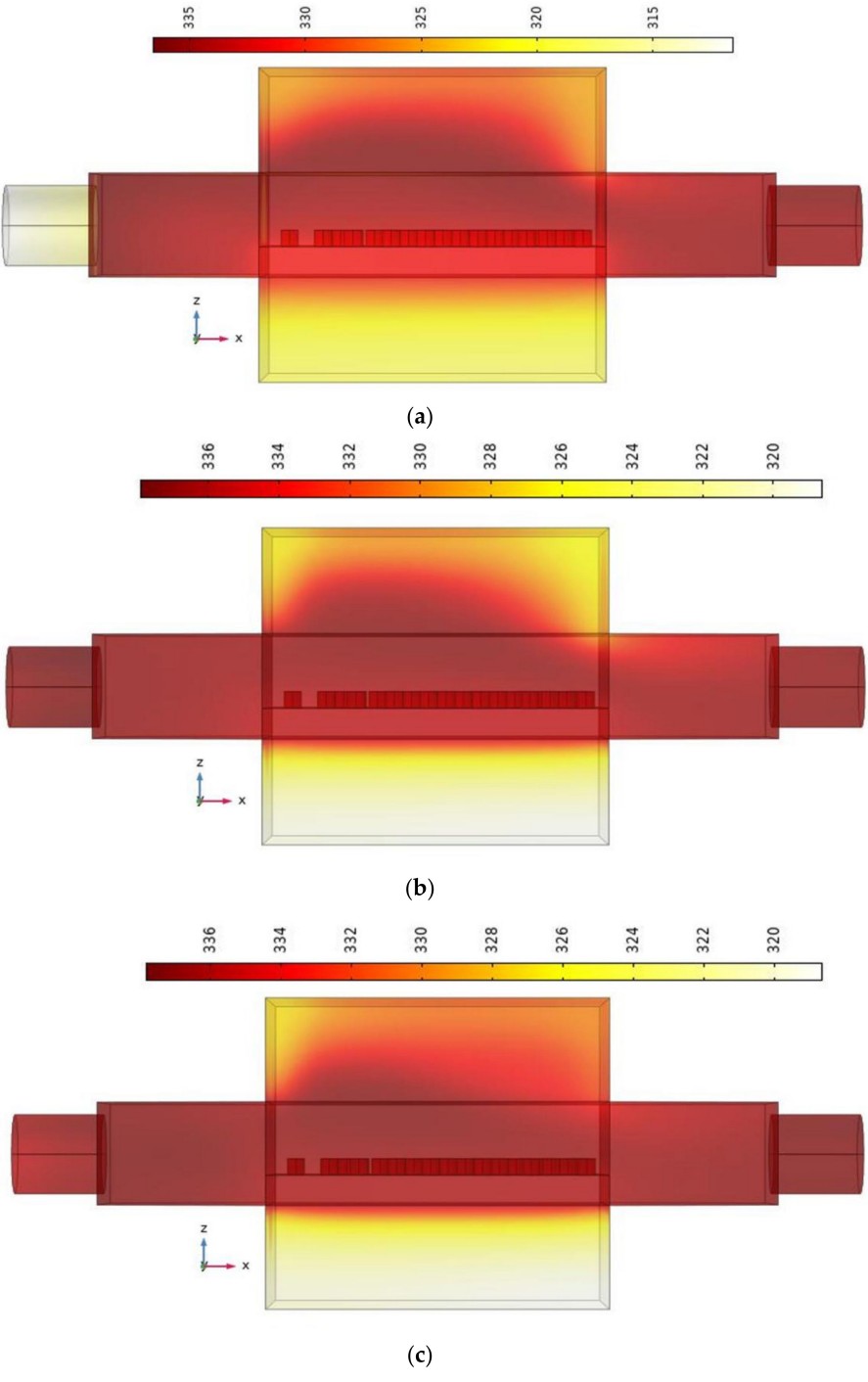

**Figure 10.** Iso surface temperature distribution at the end of the thermal storage charging process in the presence of pin-fins. (**a**) Re = 50, (**b**) Re = 100, (**c**) Re = 150.

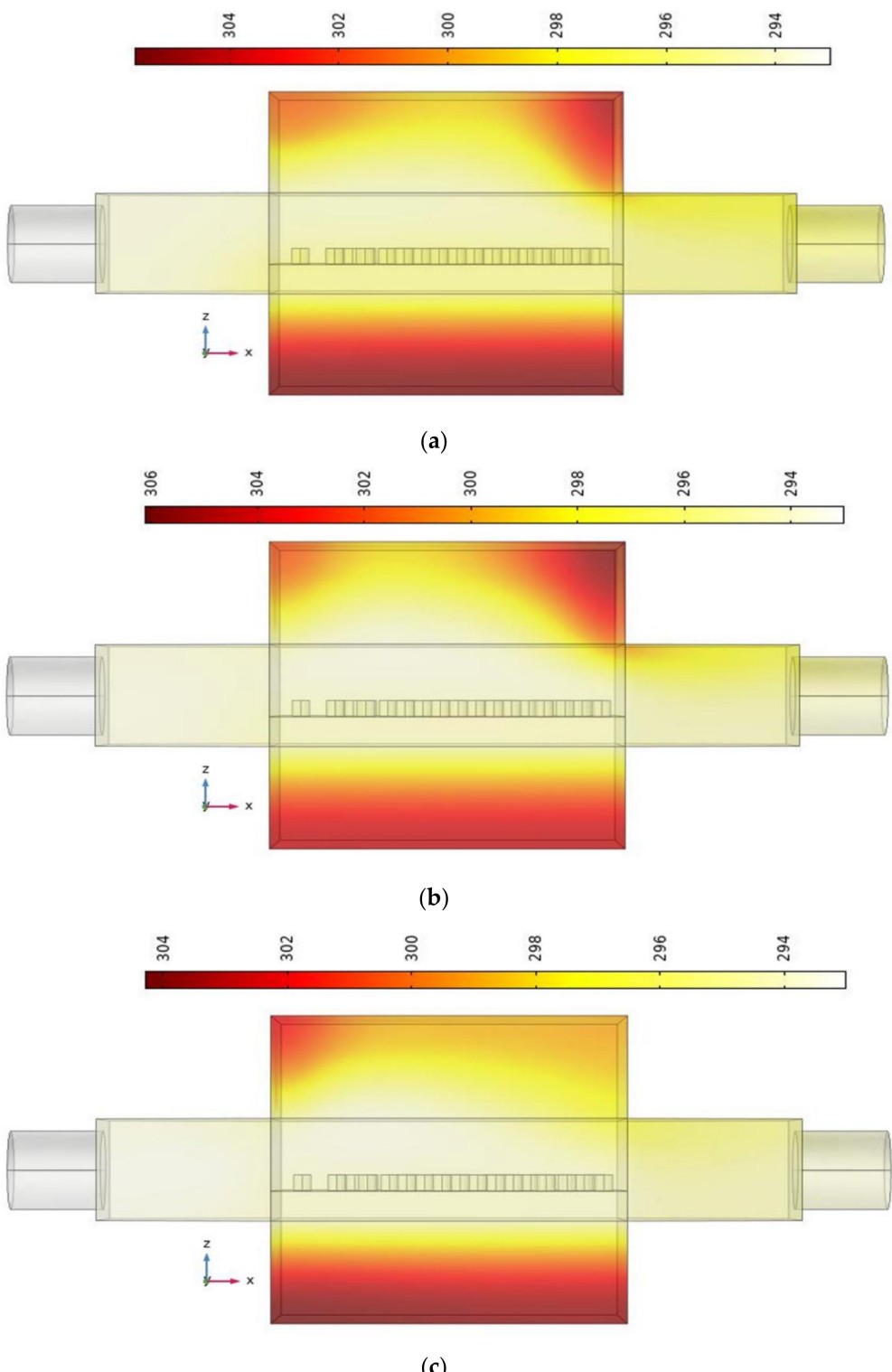

**Figure 11.** Iso surface temperature distribution at the end of the thermal storage discharging process in the presence of pin-fins (**a**) Re = 50, (**b**) Re = 100, (**c**) Re = 150.

Figure 12 shows the overall iso-surface temperature distribution of the model for the case of a Reynolds number of 50 during charging (i.e., Figure 12a) and discharging (i.e., Figure 12b). The heat trapped at the corner of the PCM is present in Figure 12b, and as the Reynolds number increases, the amount of heat trapped is further reduced, as demonstrated earlier.

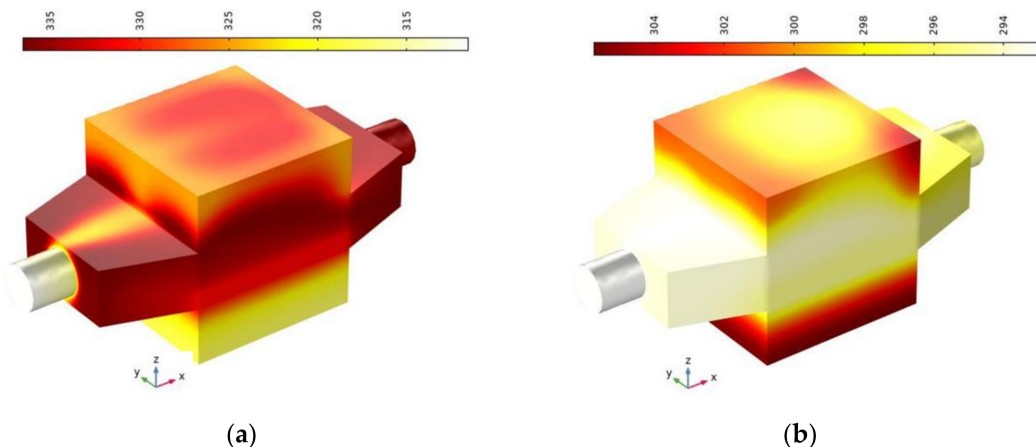

(**a**)                                                                    (**b**)

**Figure 12.** Temperature variation for the block phase change materials at t = 700 s and t = 1400 s (Re = 50).

The reason for more heat at the top of the PCM instead of the bottom PCM is investigated further by calculating the heat fluxes at the top and bottom interfaces. Figures 13 and 14 display the heat flux at the interfaces during charging and discharging. The top interface is located at the intersection between PCM and the top boundary of the flow region, and the bottom interface is located at the base of the pin-fins. This total heat flux is composed of conductive heat flux and convective heat flux. The flow rate in question corresponds to a Reynolds number of 50. During the charging process, an equal amount of heat is conducted to both interfaces. During the discharge period, the top interface releases heat faster due to the natural convection. Figure 14 shows the total heat flux, including conductive and convective heat flux. One can notice a greater heat flux for the top PCM than for the bottom interface. The reason is that the presence of pin-fins created a vigorous mixing in the flow by reflecting the heat to the top interface. This can demonstrate the reason for high energy storage in the top PCM. Implementing pin-fins can assist in transferring the heat in a fast way.

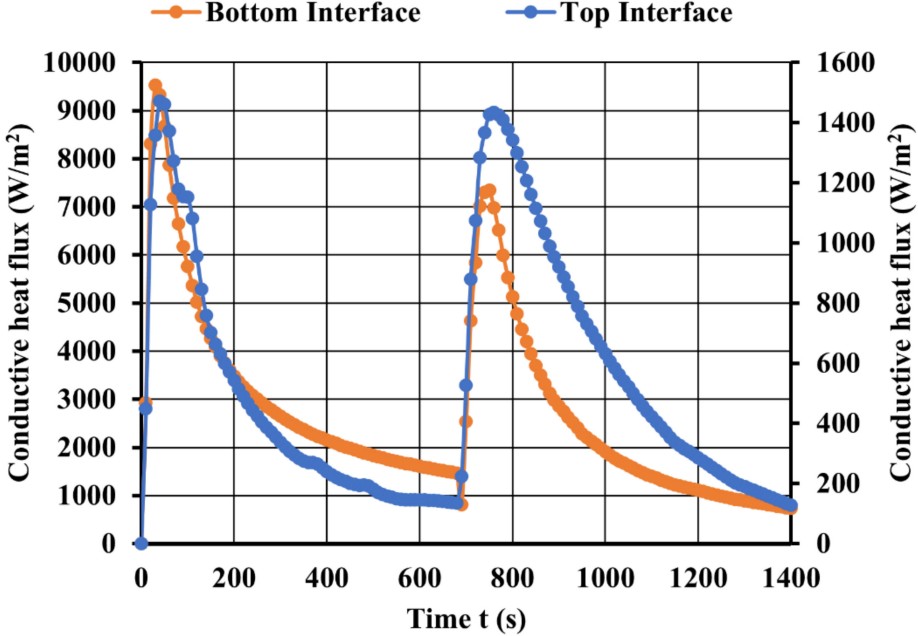

**Figure 13.** Conductive heat fluxes at the top and bottom interface (Re = 50).

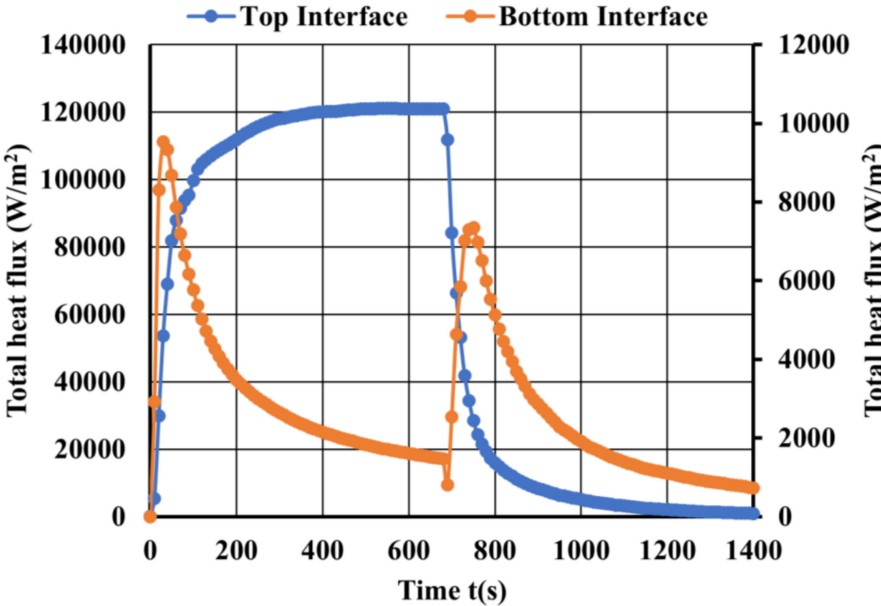

**Figure 14.** Total heat fluxes at the top and bottom interface (Re = 50).

*5.3. Effectiveness of Pin-Fin Height on the Amount of Energy Stored in the Phase Change Material*

In the previous section, we demonstrated the effectiveness of using pin-fins to enhance heat transfer toward phase change material. In the last investigation, the pin-fins height was set to 2 mm. The calculation was repeated for pin-fins height varying between 1 mm and 6 mm. All other input conditions remained identical.

Figure 15 presents the total heat stored in the phase change material by calculating Q during the charging period over the time from zero seconds to 700 s. As shown, as the pin-fins height increase, the heat transfer improves, and since the mixing is increased, a large heat flux is deflected toward the top phase change material block. What is interesting from this graph is that it appears that an optimum height is obtained equal to 5 mm. As the pin-fins height further increases, the pin-fins performance decreases. It is believed that this is due to the flow behavior.

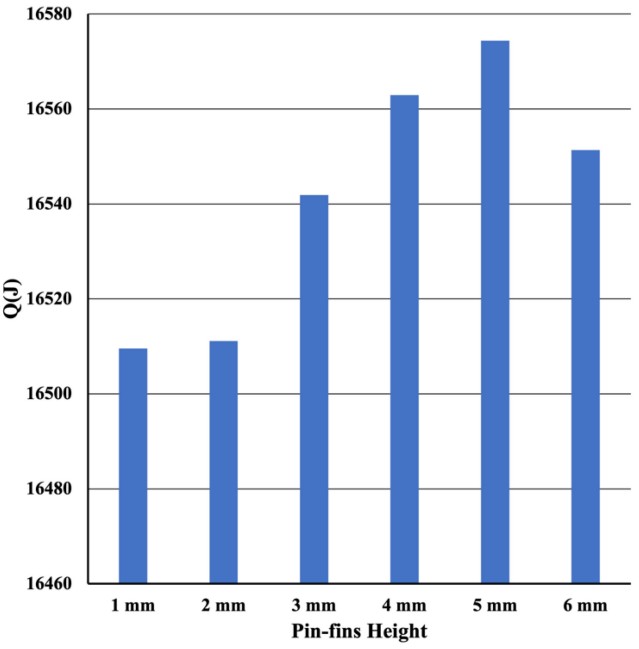

**Figure 15.** Energy stored (Q) as a function of pin-fins height (Re = 150).

To investigate this finding further, Figure 16 presents the total heat flux at the top interface during charging and discharging. It is evident that after a pin-fins height of 5 mm, the total heat flux becomes constant and occupies a smaller area as a function of time. If this flux is integrated over the charging period only, Figure 17 presents the average total heat flux. Indeed at a pin-fins height of 6 mm, the heat flux drops in magnitude as predicted by calculating the energy stored in the phase change material. As indicated earlier, most heat transfer activities occur at the top block of the phase change material, and gravity enhances this heat transfer by natural convection.

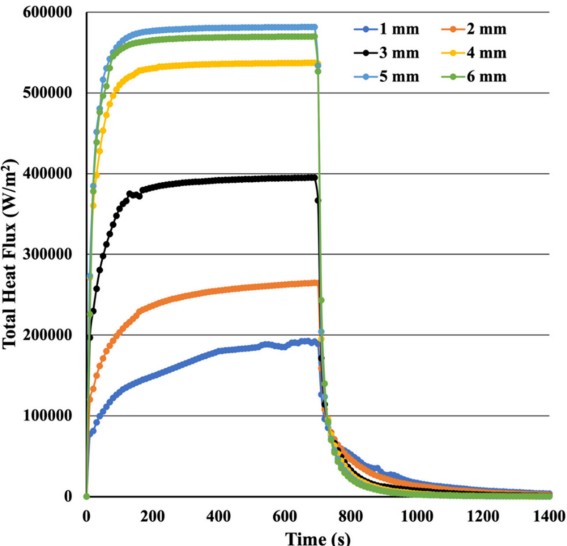

**Figure 16.** Total Heat flux during charging and discharging.

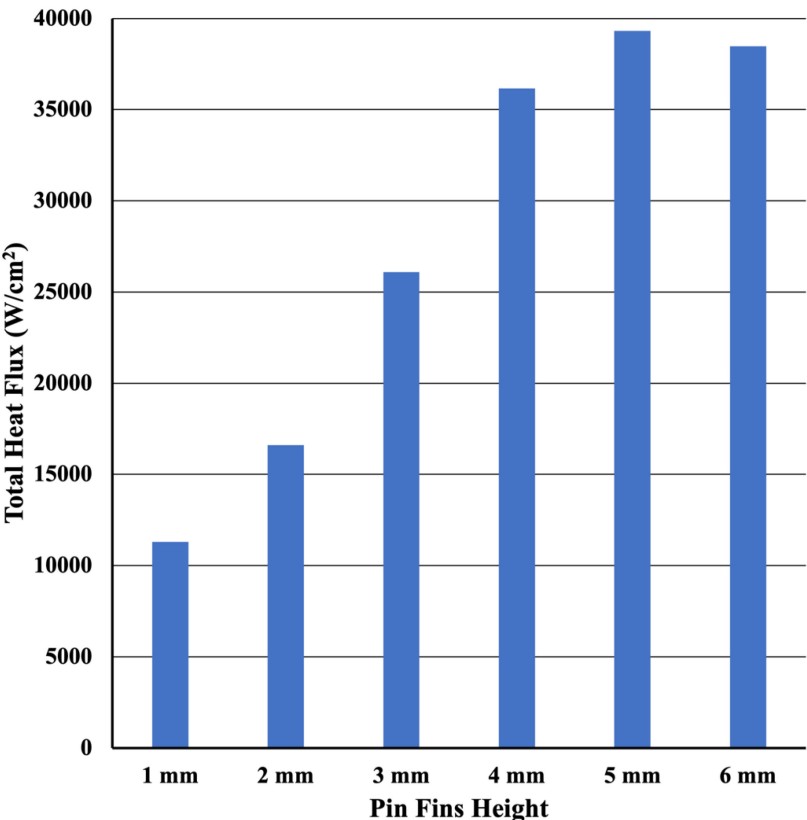

**Figure 17.** Average total heat flux at the top phase change material block.

## 6. Conclusions

Wastewater energy removal is the main topic of the present paper. The energy harvested is stored in a block of phase change material for use during the discharging process. Pin-fins are inserted in the flow path aiming at high heat transfer performance. The two-phase change material blocks were subject to natural convection for the first block and pure diffusion for the second block. Results revealed:

1. Pin-fins are the best approach to transfer heat to the phase change material;
2. The use of pin-fins created a mixing flow in the liquid region, thus increasing the convective heat flux;
3. The presence of pin-fins accelerated the heat storage capability in phase change material;
4. The height of pin-fins influences the heat transfer, and for the present configuration, a 5 mm pin-fin is the optimum performance;
5. Natural convection presence can accelerate the heat storage capability in the phase change material.

**Funding:** The National Science and Engineering Research Council, Canada (NSERC, Grant No. 158025).

**Institutional Review Board Statement:** Not applicable.

**Informed Consent Statement:** Not applicable.

**Data Availability Statement:** Not applicable.

**Conflicts of Interest:** No conflict of interest.

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
