# Peer review of "Enhanced Energy Storage Using Pin-Fins in a Thermohydraulic System in the Presence of Phase Change Material"

_fluids, doi:10.3390/fluids7110348_

Round 1

Reviewer 1 Report

The main findings are to be presented in the abstract.

Why the inlet velocities are relatively low?

The energy equation at the solid zone is to be presented.

How the heat flux continuity at the solid-fluid interface is ensured?

The boundary conditions are to be expressed mathematically.

A figure presenting the used mesh is to be presented.

The validation/verification of the numerical model should be performed by comparing with a study dealing with PCM and not a simple fluid or nanofluid. In addition, a qualitative validation/verification is to be presented.

The configuration studied in Ref 29 is not the same as yours.

 The numerical method is to be detailed.

How the phase change is treated numerically ? how the solid and liquid PCM are differentiated?

There is a sub-title ‘’ Mesh sensitivity and convergence criteria’’ but the convergence criteria is not presented.

The title of figure 2 is ‘’ Experimental and numerical setup’’ but there is no experimental setup.

The expression of Nusselt is to be added.

The 3D flow structures and temperature fields are to be added.

Author Response

Dear Colleague

Thank you for taking the time to review my paper. Your comments are greatly appreciated

Best Regards

Ziad

Reviewer 2 Report

Review of  fluids-1984357„Enhanced Energy Storage using Pin-Fins in a Thermohydraulic System in the Presence of Phase Change Material” by M.Ziad Saghir

   This study proposes a numerical model regarding the effectiveness of pin-fins to store/release energy when immersed in a phase change material. The system thermal response is investigated when a step function temperature is applied, at three different Reynolds numbers, with reported data on temperature outlet behavior in time, energy charge and discharge power, axial temperature distribution and it stime evolution in „no-fin” and „fin” configurations, isosurface temperature distributions at the beginning and at the end of one cycle. The investigation is thorough and the results are clearly presented, well-structured and support the drawn conclusions. Comparison with some experimental measurements would be useful.

 Some review comments are listed below:

 1.     In Fig. 3, in the low Reynolds region (600-1000), the numerical Nu values are somewhat different from the experimental measurements. Given the fact that the simulation is within the low Reynolds number interval, please discuss whether these differences influence your data.

2.     A short statement on the benefits of using a chevron pin arrangement from a hydrodynamics and thermal points of view, besides the reference (29), would be beneficial.

3.     The phrase „using pin-fins as a simple mechanism for heat storage” is overstated. Pin –fins, just as metal foams, can be an useful tool to enhance heat storage, rather than a mechanism.

4.     The differences between stored energy in „no-pin” and „pin” configurations, presented in Table 3, seem small. Calculation of a global heat transfer coefficient, U, or UA/V maybe would underline more the effectiveness of „pin-fin” configuration.

5.     Figure 11 a ,b delivers no extra information in comparison to Figures 9 a and 10 a.

6.     Some minor typos should be addressed, for ex. in eq(8 ), there is an extra ρ, etc.

Author Response

Dear Colleague

Thank you for taking the time to review my paper. Your commenst and suggestions are greatly appreciated

Regards

Ziad

Round 2

Reviewer 1 Report

After revision, the paper can be accepted for publication